# A multispectral 3D live organoid imaging platform to screen probes for fluorescence guided surgery

Bernadette Jeremiasse[1,2], Ravian L van Ineveld[1,2], Veerle Bok [1,2], Michiel Kleinnijenhuis[1,2], Sam de Blank [1,2], Maria Alieva[1,3], Hannah R Johnson[1,2], Esmée J van Vliet [1,2], Amber L Zeeman[1,2], Lianne M Wellens [1,2], Gerard Llibre-Palomar[1,2], Mario Barrera Román [1,2], Alessia Di Maggio[4,5], Johanna F Dekkers[1,2], Sabrina Oliveira[4,5], Alexander L Vahrmeijer[6], Jan J Molenaar[1], Marc HWA Wijnen[1], Alida FW van der Steeg[1], Ellen J Wehrens[1,2] & Anne C Rios [1,2✉]

## Abstract

**Achieving complete tumor resection is challenging and can be improved by real-time fluorescence-guided surgery with molecular-targeted probes. However, pre-clinical identification and validation of probes presents a lengthy process that is traditionally performed in animal models and further hampered by inter- and intra-tumoral heterogeneity in target expression. To screen multiple probes at patient scale, we developed a multispectral real-time 3D imaging platform that implements organoid technology to effectively model patient tumor heterogeneity and, importantly, healthy human tissue binding.**

**Keywords** Fluorescence-guided Surgery; Patient-derived Organoids; Multi-spectral 3D Imaging; Neuroblastoma; Breast Cancer
**Subject Categories** Cancer; Methods & Resources

## Introduction

Fluorescence-guided surgery (FGS) is an intra-operative imaging technique that uses a fluorescent agent and a near-infrared camera system to help surgeons visually detect tumor tissue in real-time (Mieog et al, 2022; Koller et al, 2018). FGS has been proven beneficial for enhancing radical tumor resections and could reduce complication risks (Gao et al, 2018). Molecular-targeted FGS, applying probes that bind to membrane markers overexpressed on tumor cells, is rapidly evolving (Mieog et al, 2022). The success of this approach depends on identifying targets that are specifically expressed on living tumor tissue.

Finding suitable FGS probes is challenging, due to the heterogeneity of target expression, and typically requires time-consuming individual testing using xenograft models (Pogue et al, 2018). This hampers the speed of pre-clinical identification and

poses a risk of selecting probes that recognize healthy human tissue, due to interspecies differences. For many tissue types, organoids have been developed from both healthy and cancerous human tissues (Tuveson and Clevers, 2019), making them a suitable candidate model system for FGS probe prioritization. Moreover, human organoid technology offers a unique in vitro window into patient-representative biological processes and patient-derived organoid (PDO) biobanks are already being used to prioritize quality leads in cancer drug development and study treatment efficacy in a personalized manner (Dekkers et al, 2022).

By combining organoid technology (Tuveson and Clevers, 2019; Dekkers et al, 2021) with our latest advances in multi-spectral imaging (Dekkers et al, 2022; van Ineveld et al, 2021) and large-scale data segmentation (van Ineveld et al, 2021), we developed a multispectral 3D live organoid imaging platform to identify molecular-targeted FGS probes at patient scale (Fig. 1A, Movie EV1). By applying it to both a neuroblastoma (NB) (Kholosy et al, 2021) (Table EV1) and a breast cancer (BC) PDO biobank (Dekkers et al, 2021; Sachs et al, 2018) (Table EV2), we demonstrate broad applicability for pediatric and adult oncology. In addition, our BC PDO biobank contains a metastasis-derived line (Dekkers et al, 2021) (169M; Table EV2), thereby, illustrating the potential of our platform to define molecular probes relevant for metastasis detection. Importantly, we highlight the discovery power of our platform by identifying new targets to specifically light up NB tissue, as well as potential probe combinations for BC. Finally, validation and comparison with in vivo data demonstrates a critical need to model healthy human tissue in near-proximity to the surgery location that can be achieved by our organoid-based approach.

## Results and discussion

Our platform can perform 7-color 3D live organoid imaging in single-scan acquisition by using spectral imaging and linear unmixing, thereby screening up to six FGS probes simultaneously

[1]Princess Máxima Center for Pediatric Oncology, Utrecht, The Netherlands. [2]Oncode Institute, Utrecht, The Netherlands. [3]Instituto de Investigaciones Biomedicas Sols-Morreale (IIBM), CSIC-UAM, Madrid, Spain. [4]Pharmaceutics, Department of Pharmaceutical Sciences, Utrecht Institute for Pharmaceutical Sciences, Utrecht University, 3584 CG Utrecht, The Netherlands. [5]Cell Biology, Neurobiology and Biophysics, Department of Biology, Science Faculty, Utrecht University, 3584 CH Utrecht, The Netherlands. [6]Department of Surgery, Leiden University Medical Center, Leiden, The Netherlands. ✉E-mail: a.c.rios@prinsesmaximacentrum.nl

(Figs. 1B,C and EV1A,B) alongside a general cellular marker. Upfront acquisition of reference emission spectra for all fluorophores precludes the need for individual fluorophore control samples, advancing the speed and ease of sample preparation and image acquisition (van Ineveld et al, 2021) (Fig. EV2A).

Targets for probe screening were selected based on published RNA sequencing datasets (Fig. EV2B) and literature review (Table EV3). Selected targets included three common targets; GD2, NCAM1, and L1CAM, as well as additional targets per tumor type; ALCAM and THY1 for NB and EGFR, EPCAM, and HER2 for BC. In the absence of readily available directly conjugated fluorescent probes, probes were generated by fluorescent conjugation of specific antibodies against the identified targets (Table EV4). To quantify probe binding and compare it between the different PDO lines (Fig. 1B,C), we implemented an optimized module in our recently developed STAPL-3D pipeline (van Ineveld et al, 2021) (Fig. EV3A,B), segmenting over 11,700 individual organoids across 21 lines analyzed. For each probe selected for the different PDO biobanks, we were able to quantify the percentage of positive organoids and their relative intensity (Figs. 1D,E and EV3C,D).

The resulting comprehensive dataset allows to map probe binding intensity in uniform manifold approximation and projection (UMAP) space (Fig. EV3E,F) and clustering based on spatial target distribution shows that the majority of PDO lines predominantly separate into distinct clusters (Fig. 2A,B), especially for heterogeneous breast cancer (Sachs et al, 2018). This illustrates that our platform captures the critical influence of inter-patient tumoral heterogeneity on FGS probe labeling efficacy. Moreover, even intra-tumoral heterogeneity in probe binding was detected, reflected by the presence of multiple clusters per PDO line (Fig. 2C,D), as well as distinct probe expression patterns between single organoids from the same line in the fluorescent imaging data (Fig. 2E).

To validate the results of our in vitro screening platform, we tested optimized dosages of anti-GD2 and anti-L1CAM (Fig. EV4A–D), already available as immunotherapy or in the process of obtaining GMP approval, respectively (Yu et al, 2010; Künkele et al, 2017) (Table EV3), custom labeled with a commonly used FGS infra-red dye; IRDye800CW (Wellens et al, 2020). We established mouse xenografts (Fig. 3A) with tumors originating from three different NB PDO lines. These lines were selected for their differential binding of GD2 probe in our screening assay (Fig. 3B), allowing us to in vivo validate inter-tumoral heterogeneity in probe binding quantified in vitro. The results show that in vitro and in vivo tumor-to-background ratios (TBRs) for GD2 binding are not different and a similar relative pattern between the PDO lines can be observed (Fig. 3C). This consistency demonstrates the in vivo relevance of in vitro obtained results with our screening platform.

Importantly, TBRs for anti-L1CAM were well above the clinical cut-off value of 1.5 in vivo (Tummers et al, 2018), a cut-off that is generally considered acceptable for use in FGS (Galema et al, 2022) (Fig. 3D). This sufficiently high in vivo TBR results from discriminative binding to human tumor tissue compared to healthy mice organs when using a human-specific antibody (Fig. EV4E–H). However, this is not the case in vitro (Fig. 3D), where we were able to define the TBR on actual binding of the probes to human healthy tissue that typically surrounds the NB surgery site, modeled with kidney organoids (Calandrini et al, 2020; Schutgens et al, 2019)

(Fig. 1B). This resulted not only in L1CAM no longer reaching the desired cut-off value of 1.5, but also in borderline values for ALCAM, again underscoring the importance of relevant healthy tissue modeling (Fig. 3E). In line with our previous finding (Wellens et al, 2020), this method confirmed anti-GD2 as a promising probe for FGS in NB, but, importantly, also showed that both NCAM1 and THY1 that have not been suggested for FGS before, might even have higher discriminative power for some patients (Fig. 3E). TBR analysis of our selected targets for BC, overall revealed a lower discriminative power. EPCAM, previously suggested for FGS in BC (Boogerd et al, 2019) and indeed largely expressed across PDO lines (Fig. 1E), only showed an acceptable TBR above 1.5 for three out of nine donors, due to healthy tissue binding (Fig. 3F). In addition, HER2, under pre-clinical development as a BC FGS probe (Deken et al, 2019), only reached a sufficient TBR for lines 100T and 10T (Fig. 3F), also identified to have the highest immunohistochemistry (IHC) score for HER2 over-expression (Table EV2). Thus, in line what has been suggested in literature (Kedrzycki et al, 2022), our results indicate that for more heterogenous tumor types like BC, probe combinations might be required to reach sufficient coverage across patients. Exploiting the scalability of our platform, we used it to screen 6 single probes and 57 probe combinations in total (Figs. 3G and EV5, Table EV5), showing that specific probe combinations could increase the total percentage of BC organoid coverage, with EPCAM-HER2 and NCAM1-EPCAM-HER2 presenting the most feasible combinations with a limited number of probes (Fig. 3G). While for the majority of PDO lines, these combinations increased organoid coverage, the extent of this varied between approximately 40 to 70%, depending on the specific line. For difficult to target triple-negative 36T coverage remained low, whereas 13T provides a clear example of the benefit of adding NCAM1 to the probe mix (Fig. EV5). Together, this addresses a current clinical challenge (Kedrzycki et al, 2022), by identifying two sets of probe combinations for further follow-up in BC. Moreover, it underscores the importance of modeling tumor heterogeneity in target expression and large-scale multiplex screening to identify FGS probes and probe combinations for heterogeneous tumor types.

In sum, here we developed an organoid-based multiplex imaging platform to screen for FGS probes. We demonstrated its potential for finding new probes and promising probe combinations, as well as preventing false positive results, due to healthy tissue binding. Showcased on two PDO biobanks, we consider this platform a versatile and broadly applicable discovery tool, especially given the widespread availability of organoid biobanks from healthy and tumor tissues and their continued development from additional tissue sources (Tuveson and Clevers, 2019). By providing crucial information on labeling efficacy, tumor heterogeneity in spatial distribution, and discrimination between tumor-specific and healthy-background signal in a human setting, it can complement traditional animal models currently in use for FGS probe assessment. Main advantages of our organoid-based in vitro platform stem from its throughput character and accessibility. This enables modeling of healthy human tissue binding that cannot be captured in mouse xenograft models. In addition, the ability to screen multiple probes simultaneously offers an important benefit, not only regarding timeline of development, but also for more heterogenous tumor types that might require probe combinations for tumor detection across patients. Results from our BC PDO

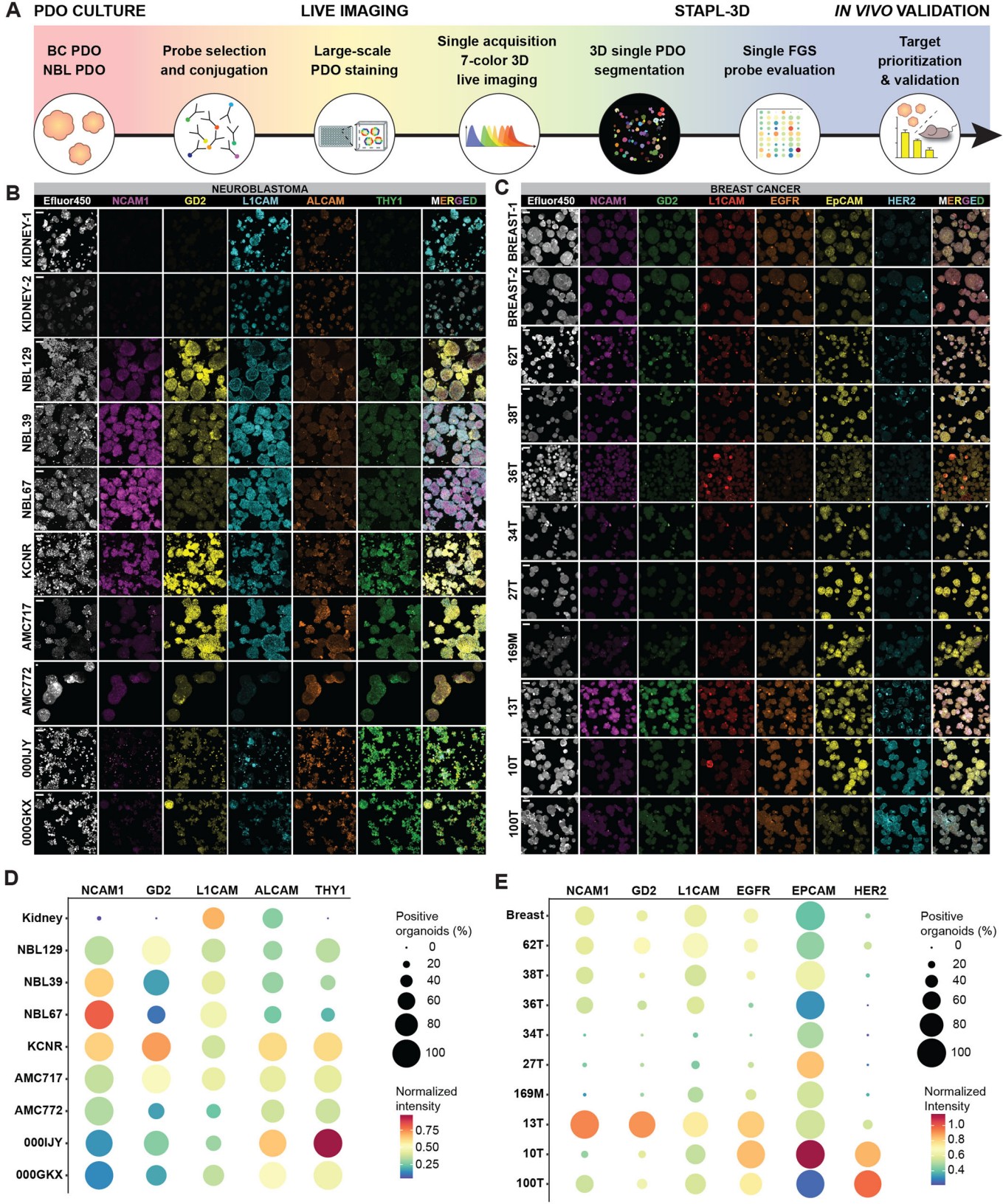

Figure 1. Organoid-based multispectral 3D imaging platform to screen FGS probes.

(A) Schematic representation of the screening platform, including 7-color 3D imaging and STAPL-3D single organoid segmentation and quantification for target prioritization. (B, C) Representative single channel and merged 3D multispectral images of the segmented data, showing heterogeneous expression of selected FGS probes between PDO lines. Scale bars 50 µm. (D, E) Quantification of percentage positively stained organoids (dot size) and normalized fluorescent intensity (blue-to-red color gradient) for the two PDO biobanks and healthy tissue control organoids. (B–E) $n = 3$ (NB) and $n = 4$ (BC) independent experiments. Source data are available online for this figure.

biobank provide proof-of-concept that a combination of probes can extend the number of patients that would benefit from an FGS approach. Another tumor indication where this would be valuable to investigate is brain cancer. For now, 5-aminolevulinic acid (5-ALA) is used in this context, and while very effective for newly diagnosed high-grade glioma (HGG), specificity is much lower for low-grade glioma or recurrent HGG (McCracken et al, 2022; Dadario et al, 2021). Therefore, it would be worth investigating if (a combination of) specifically targeted probes could offer broader detection across brain tumors, especially since both healthy and tumor brain organoid models are available to evaluate this (Abdullah et al, 2021; Jacob et al, 2020; Lago et al, 2023; Lancaster et al, 2013; Hendriks et al, 2024). However, blood-brain-barrier crossing properties of identified probes will have to be considered as well (Bergmann et al, 2018). In general, clinical benefits of probes identified with our organoid screening platform will depend on a multitude of factors, including tumor characteristics, probe properties, disease advancement, and patient responses and, thereby, will require further testing in in-patient clinical trials (Mieog et al, 2022).

After pre-selection of probes on PDOs, the same 3D multispectral imaging and analytical pipeline could be performed on biopsy material to define the most promising (combination of) probes for precision medicine. Moreover, implementation can reach beyond FGS, as a similar approach could be envisioned to live screen for tumor overexpressed membrane markers for targeted therapy or cellular immunotherapy.

# Methods

## Ethics

All NB PDO lines were received from the biobank of the Princess Máxima Center (PMCLAB2019.037) and BC PDO lines from a biobank through the Hubrecht Organoid Technology (HUB; huborganoids.nl) with informed consent from all donors. Approval from the medical ethical committee of the UMC Utrecht (NedMec) ensured compliance with the Dutch Medical Research Involving Human Subjects Act. Generation of normal breast organoids from milk was approved by the Clinical Research Committee of the Princess Máxima Center. All animal experiments were approved by the Animal Welfare Committee AVD3990020173067 of the Princess Máxima Center and carried out in compliance with international ethical regulations. In addition, all experiments conformed to the principles set out in the WMA Declaration of Helsinki and the Department of Health and Human Services Belmont Report.

## Patient-derived organoid culture

Patient-derived NB organoids; NBL39, NBL67, NBL129, AMC717, AMC772, 000IJY, and 000GKX (Table EV1) were cultured as described previously (Kholosy et al, 2021). Briefly, Dulbecco's modified Eagle's medium (DMEM)-GlutaMAX containing low glucose was supplemented with 20% Ham's F-12 Nutrient Mixture, B-27 Supplement minus vitamin A, N-2 supplement, 100 IU/ml penicillin and 100 µg/ml streptomycin (pen/strep) (all Thermo Fisher), 20 ng/ml epidermal growth factor (EGF), 40 ng/ml fibroblast growth factor-basic (FGF-2), 20 ng/ml insulin-like growth factor (IGF-1), 10 ng/ml platelet-derived growth factor AA (PDGF-AA) and 10 ng/ml platelet-derived growth factor BB (PDGF-BB) (all Peprotech). Organoids grew mostly in suspension in T75 cell culture flasks (Cellstar). Medium was refreshed every week and organoids were passaged 1:2–1:16 every 7–14 days, using TrypLE Express (Thermo-Fisher) in case of partial adherence. Two healthy kidney control organoid lines were maintained by the Drost group (Princess Máxima Center, Utrecht, Netherlands) (Schutgens et al, 2019; Calandrini et al, 2020) and kindly provided prior to imaging.

Patient-derived BC organoids; 10T, 13T, 27T, 34T, 36T, 38T, 62T, 100T, and 169M (Table EV2) and two normal breast organoid lines were cultured as described previously in 12-well suspension plates (Greiner Bio-One) seeded in basement membrane extract (BME; Cultrex) (Dekkers et al, 2021; Sachs et al, 2018). Briefly, advanced DMEM/F12 was supplemented with pen/strep, 10 mM HEPES, GlutaMAX (F12+++), 1× B27 (all Thermo Fisher), 1.25 mM N-acetyl-l-cysteine (Sigma-Aldrich), 10 mM nicotinamide (Sigma-Aldrich), 5 µM Y-27632 (Abmole), 5 nM Heregulin β-1 (Peprotech), 500 nM A83-01 (Tocris), 5 ng/ml epidermal growth factor (Peprotech), 20 ng/ml human fibroblast growth factor (FGF)-10 (Peprotech), 10% Noggin-conditioned medium, 10% Rspo1-conditioned medium, and 0.1 mg/ml primocin (Thermo Fisher); and additionally with 1 µM SB202190 (Sigma-Aldrich) and 5 ng/ml FGF-7 (Peprotech) for PDO propagation (type 1 culture medium (Dekkers et al, 2021)), or with 20% Wnt3a-conditioned medium, 0.5 µg/ml hydrocortisone (Sigma-Aldrich), 100 µM β-estradiol (Sigma-Aldrich), and 10 mM forskolin (Sigma-Aldrich) for normal organoid propagation (type 2 culture medium (Dekkers et al, 2021)). Culture medium was refreshed every 2–3 days and organoids were passaged 1:2–1:6 every 7–21 days using TrypLE Express (ThermoFisher). All organoid lines were cultured in a humidified incubator at 37 °C and 5% $CO_2$ and were free of Mycoplasma species. Experiments were performed with organoids from passages 5–30 after establishment.

## SMS-KCNR cell culture

The human male neuroblastoma cell line SMS-KCNR (KCNR) (Reynolds et al, 1986) was authenticated upon acquisition using STR profiling and cultured in Dulbecco's Modified Eagle's Medium (DMEM)-GlutaMAX containing low glucose, 10% Fetal Bovine Serum (FBS), 1% MEM Non-Essential Amino Acids, 1x GlutaMAX, and pen/strep (all Thermo Fisher). Culture medium was refreshed every week and cells were passaged 1:2–1:6 every 7–14 days using TrypLE Express (ThermoFisher). Cells were cultured in a humidified incubator at 37 °C and 5% $CO_2$ and were free of Mycoplasma species.

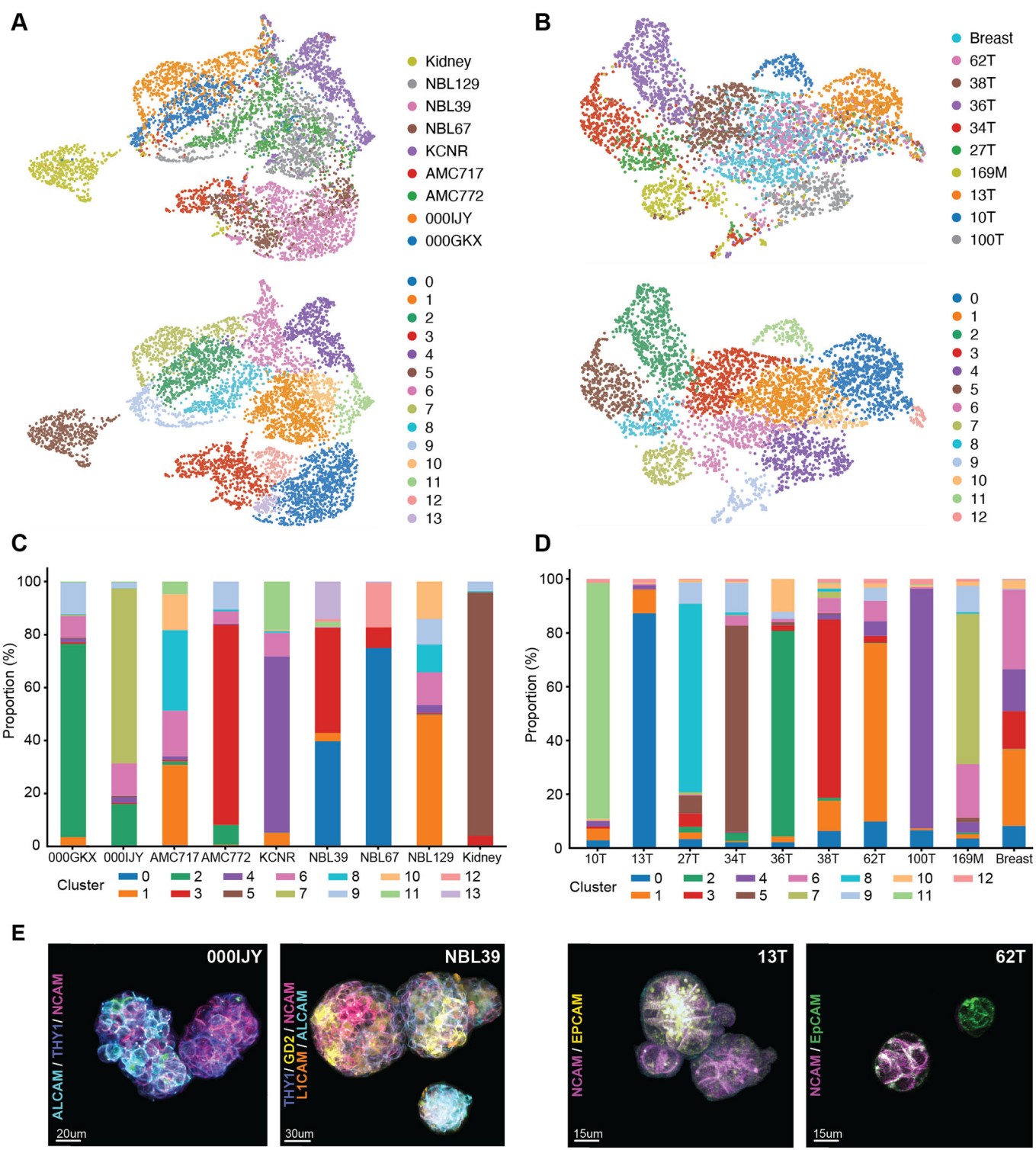

**Figure 2. Inter-patient and intra-tumoral heterogeneity in probe binding.**

(A, B) UMAP clustering based on spatial target distribution for both the NB (A) and BC (B) PDO biobank, $n = 3$ independent experiments. (C, D), Stacked bar graphs representing the proportion of each cluster per PDO line of the NB (C) and BC (D) PDO biobank, $n = 3$ independent experiments. (E), Representative 3D multispectral images showing heterogeneous binding of selected FGS probes on individual organoids within the same PDO line. Scale bars 20 μm for representative image of 000IJY, 30 μm for NBL39 and 15 μm for representative images of 13T and 62T.

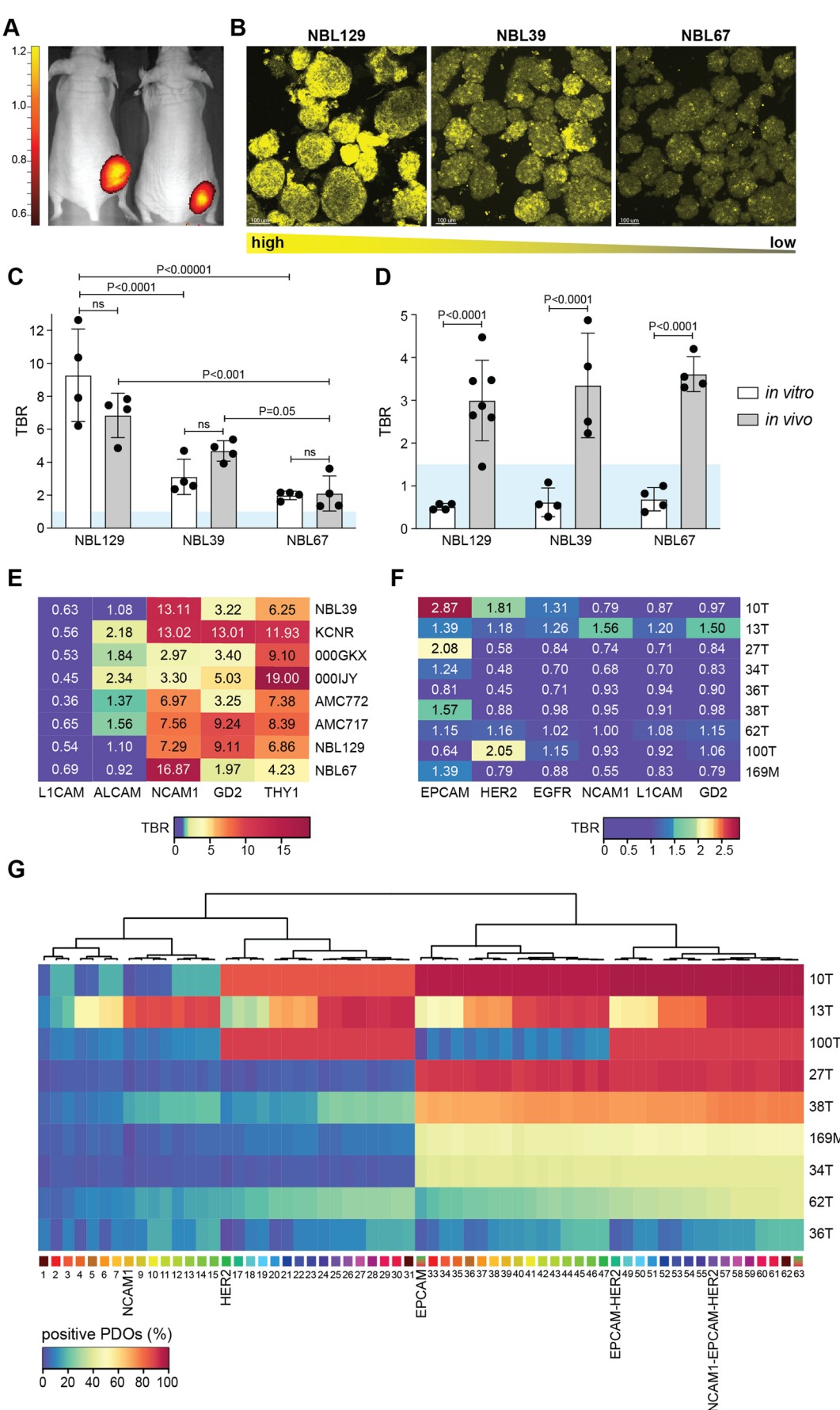

**Figure 3.  In vivo validation and probe combination screening.**

(A) Representative in vivo anti-L1CAM-IRDye800CW3D detection images of xenografted mice bearing subcutaneous PDO NBL67 derived tumors. (B) Representative images of in vitro GD2 expression on the NBL129, NBL39, and NBL67 PDO lines, re-used from Fig. 1A. Scale bars 100 μm. (C) In vitro (white bars) and in vivo (gray bars) TBRs of GD2. Individual data points shown, and bars depict mean TBR + SD. Blue area indicates common TBR cut-off value of 1.5. Comparison between different NBL lines vitro: NBL129 versus NBL39 adjusted $P$-value: 2.90E−05; NBL129 versus NBL67 adjusted $P$-value: 3.00E−06; NBL39 versus NBL67 adjusted $P$-value: 0.52, two-way ANOVA with Sidáks multiple comparisons test. Comparison between different NBL lines vivo: NBL129 versus NBL39 adjusted $P$-value: 0.11; NBL129 versus NBL67 adjusted $P$-value: 5.51E−04; NBL39 versus NBL67 adjusted $P$-value: 0.05, two-way ANOVA with Sidáks multiple comparisons test. Comparison between in vitro and in vivo TBR: NBL129 adjusted $P$-value: 0.08; NBL39 adjusted $P$-value: 0.36; NBL67 adjusted $P$-value: 1.00, two-way ANOVA with Sidáks multiple comparisons test. $n = 3$ independent in vitro experiments, and $n = 4$ mice in vivo per PDO line. (D) In vitro (white bars) and in vivo (gray bars) TBRs of L1CAM. Individual data points shown, and bars depict mean TBR + SD. Blue area indicates common TBR cut-off value of 1.5. Comparison between in vitro and in vivo TBR: NBL129 adjusted $P$-value: 5.50E−05; NBL39 adjusted $P$-value: 7.48E−05; NBL67 adjusted $P$-value: 3.15E−05, two-way ANOVA with Sidáks multiple comparisons test. $n = 3$ independent in vitro experiments and $n = 4$-7 mice in vivo per PDO line. (E, F) In vitro TBR (blue-to-red color gradient) for all tested probes and PDO lines for NB (E) and BC (F). $n = 3$ (NB) and $n = 4$ (BC) independent in vitro experiments. (G) Percentage of BC organoids (blue-to-red color gradient) positive for individual probes and probe combinations. Results ordered through unsupervised clustering. $n = 4$ independent in vitro experiments. Source data are available online for this figure.

## Fluorescence-guided surgery probe selection

Five probes potentially suitable for fluorescence-guided surgery (FGS) of NB were selected by screening for overexpressed cell-membrane targets in three NB RNA sequencing datasets (Lastowska, primary tumor $n = 30$ (Łastowska et al, 2007); Versteeg, post-treatment tumor $n = 139$ (Bandino et al, 2014); and Hiyama, primary tumor $n = 50$ (Ohtaki et al, 2010)) and three healthy control (Mas, healthy kidney $n = 192$ (Archer et al, 2022); Rainey, adrenal gland $n = 15$ (Ye et al, 2007); and Various, adrenal gland $n = 13$) RNA sequencing datasets, using the Megasampler tool of the R2 Genomics Analysis and Visualization Platform (http://r2.amc.nl) (Fig. EV2B), after which normal to low expression of the respective proteins in healthy tissue was verified in the human protein atlas (https://www.proteinatlas.org). Healthy control datasets were selected based on the anatomical location of the primary tumor, which is in most cases adjacent to the kidney and adrenal gland. Available antibodies targeting these cell-membrane receptors were selected (Table EV4). Three of these targets (NCAM-1, L1CAM and GD2) were also found of interest for FGS of BC based on R2 RNA sequencing datasets of BC and healthy breast (5 datasets: Thomssen, triple-negative biopsies $n = 124$ (Hartung et al, 2021); Prat, HER2+ biopsies $n = 156$ (Prat et al, 2014); Brown, triple-negative primary tumors $n = 198$ (Burstein et al, 2015; den Hollander et al, 2016); Meijers-Heijboer, familial breast cancer tumors $n = 155$ (Massink et al, 2015); Knudson, healthy breast $n = 32$ (Bellacosa et al, 2010) and literature review (Orsi et al, 2017; Doberstein et al, 2014; Taouk et al, 2019) (Fig. EV2B). Three additional targets for which antibodies are currently used in patients, were added to this screening panel (EpCAM, EGFR, and HER2) (Tables EV3 and EV4).

## Fluorescent probe conjugation

All fluorescently labeled probes are detailed in Table EV4. GD2, L1CAM, and NCAM-1 were conjugated with several fluorophores as described below. Chimeric monoclonal antibody against GD2 (Dinutuximab-beta, Qarziba, USA) was conjugated to Alexa Fluor™ NHS ester 514 (Invitrogen) and IRDye800CW (LI-COR Biosciences). Chimeric antibody L1CAM clone 198.5 was conjugated to Alexa Fluor™ NHS ester 555 (Invitrogen) and IRDye800CW (LI-COR Biosciences). Recombinant monoclonal nanobody NCAM-1 (QVQ), provided with a C-terminal C-Direct tag was conjugated to Alexa Fluor™ 594 C5 Maleimide (Invitrogen, A10256), according to the instructions of the manufacturer. Nanobodies binding specifically to EGFR or HER2 were produced as described previously (Deken et al, 2020; Oliveira et al, 2012) and conjugated with Alexa Fluor™ NHS ester 647 and Alexa Fluor™ NHS ester 633 (Invitrogen), respectively. The Alexa Fluor (AF) NHS esters were dissolved in anhydrous dimethylsulfoxide (DMSO) (Invitrogen, D12345), and the reaction was carried out in 0.5 M Hepes buffer (Gibco, 15630-056) pH 8.0, at room temperature for 2 h. 0.1 M Tris was added to quench the reaction. The antibody-fluorophore conjugate was purified twice using a gel filtration column (Zeba Spin Desalting Column, 40 MWKO). The degree of labeling (DoL) was calculated by measuring the protein concentration and fluorophore concentration using the NanoDrop™ One (ThermoFisher). A degree of labeling (DoL) around 1–1.5 was considered successful, as this is generally recommended for FGS probes (Rijpkema et al, 2015).

## Live PDO probe labeling

We performed 7-color live labeling using fluorescently labeled primary antibodies or nanobodies against the proposed FGS targets (Table EV4) and a general marker eFluor™450 (ThermoFisher) that binds to all cellular proteins containing primary amines. TrypLE was used to retrieve NB PDOs and KCNR cells from culture, if adherent, and for BC PDOs to dissolve BME. Organoids were transferred to a 15 mL tube containing cold adDMEM/F12, supplemented with 1x GlutaMAX, 1 mM Hepes, 20% Ham's F-12 Nutrient Mixture, 100 IU/ml penicillin and 100 μg/ml streptomycin (F12+++) (all Thermo Fisher). Organoids were pelleted at 30 g for 5 min at 4 °C to minimize the number of single cells, resuspended in 1 mL eFluor-450 (1:3000, PBS) (Thermo Fisher) and stained for 15 min in the incubator at 37 °C and 5% $CO_2$. 8 ml growth medium was added, and organoids were pelleted at $50 \times g$ for 5 min at 4 °C, resuspended in 50 μL FluoroBrite™ DMEM (Thermo Fisher) and transferred to a 1.5 mL Eppendorf tube for antibody labeling. For NB, seven PDO NB cultures, two healthy kidney organoid cultures and the KCNR cells were stained with the selected antibodies for NB (Table EV4). For BC, nine BC PDO cultures and two normal breast organoid cultures were stained with the selected antibodies for BC (Table EV4). Organoids were stained by adding 50 μL of the antibody mix and placed in a dark incubator at 37 °C and 5% $CO_2$ for 45 min. After incubation, tubes were washed by adding DMEM (NB) or F12+++ (BC) and centrifuged for 5 min at $50 \times g$, 4 °C.

## Multispectral 3D confocal imaging

After labeling, PDOs were transferred to a 96-well sensoplate microplate (Greiner BIO-ONE). Imaging was performed on a confocal microscope using $20 \times 0.8$ NA objective (Zeiss LSM880) with zoom set to 1.0 and digitized in 16 bits per voxel with voxel size $(0.415 \times 0.415 \times 0.9)$. The Online Fingerprinting mode for multispectral imaging was used, which permits the selection of reference spectra together with the excitation settings, allowing an immediate display of the unmixed images while scanning. 3D images were rendered using Imaris software (version 10.0, Bitplane). Organoids were segmented for analysis, and intensity measurements of the channels within the organoid masks were conducted. To enhance clarity, all figures exclusively depict the voxels containing organoids, thereby minimizing interference from debris and background fluorescence.

## Spectral library acquisition

To use the Online Fingerprinting mode, the positive control KCNR cell line for NB, and PDO line 13T for, BC, were used for the acquisition of the seven reference spectra (Fig. EV2A). Single immunofluorescent stainings with probe concentrations similar to the experimental setting were performed (Table EV4), and lambda stack images acquired. Unmixing was performed using the Auto Find function in Zeiss Zen Black Software (v2.3 SP1 FP1) and reference spectra saved. Linear unmixing accuracy was assessed by unmixing 32 channel lambda stacks acquired on the positive controls for NB and BC (respectively, KCNR and 13T), as described previously (van Ineveld et al, 2021) (Zeiss ZEN Blue).

## Organoid segmentation and fluorescence quantification using STAPL-3D

To segment individual organoids from the image stacks, we used STAPL3D with 3D blob detection using the Laplacian of Gaussian algorithm (https://scikit-image.org/docs/stable/api/skimage.feature.html#skimage.feature.blob_log) as its core component. For each stack, the raw data was averaged over all channels to create a mean image that covered all organoids. A clipping mask was generated by thresholding this mean image ($I > 65000$) to handle voxels that reached the intensity at the upper bound of the data range. An organoid mask was created by smoothing the mean image with a Gaussian kernel of 5 voxels and thresholding with the Otsu threshold calculated from the data within the 0–99th percentile range.

Segmentation was performed at lower resolution than the acquired data to reduce computational costs and the influence of noise and artefacts. Given the size of the organoids, these down-sampled images had ample resolution for adequate segmentation. The mean image was down-sampled in-plane by a factor 5. Similar to the full resolution, an organoid mask is produced by smoothing the down-sampled mean image with a Gaussian kernel of 1 voxel and thresholding at the Otsu threshold calculated from the 0–99th percentile range. Additionally, patches of the organoid mask smaller than 100 voxels were removed from the mask, followed by slicewise 2D hole-filling, 3D binary dilation by 3 voxels, and another round of 2D hole-filling.

The Euclidian distance transform (edt) of the organoid mask was calculated and used to detect 3D blobs at the scale of the

organoids (skimage.blob_log parameters: num_sigma = 10, threshold = 1.0, overlap = 0.5, log_scale = False, threshold_rel = None). The minimal and maximal scale varied over organoid lines to reflect differences in organoid size. For this, organoid lines were manually assigned to one of three groups: small (min_sigma = 5 μm, max_sigma = 9 μm), medium (min_sigma = 7 μm, max_sigma = 11 μm), and large (min_sigma = 9 μm, max_sigma = 17 μm). The centerpoints of the detected blobs were dilated by 7 voxels and taken as seed points for a watershed fill of the edt to fill the organoid mask. In postprocessing, organoids smaller than 5000 μm$^3$ were removed. The remaining organoids were upsampled to the original resolution [by expanding the segments by 4 μm, upsampling to the original resolution, and removing voxels outside the full-resolution organoid mask]. Finally, organoids presenting with saturated pixels were removed from the analysis. For each segmented organoid, the mean intensity of the signal was calculated for each channel.

## UMAP clustering

Uniform manifold approximation and projection (UMAP) was implemented using the Scanpy module with default parameters using the respective mean fluorescent intensities for each probe in individual organoids, normalized for the experimental day and probe. Leiden clustering (Traag et al, 2019) was used to form clusters based on these intensities, with a resolution of 0.40 and 0.45 for BC and NB, respectively.

## Analysis of fluorescent mean intensities and in vitro TBR

The cut-off for a positive stained organoid for each probe was based on the raw imaging data by three independent annotators (Fig. EV3C,D) and the mean fluorescent intensities, as determined using STAPL-3D, were visualized in the space of the original data and the cut-off value was subtracted. Percentages of positively stained organoids for each probe and PDO line were calculated by dividing the total number of positively stained organoids by the total number of imaged organoids. Fluorescent intensities were normalized for the experimental day and tested probe by dividing the raw intensities by the 95% percentile. The TBR of each probe for each PDO line was calculated by dividing the mean fluorescent intensity of all single organoids of every PDO line by the mean fluorescent intensity of the healthy control organoid lines. Tumor coverage for single and combined probes was calculated. For this, a binary parameter of positivity was generated for every probe. Single organoids were counted as positive for a specific probe when the mean fluorescent intensity was at least one standard deviation above the mean fluorescence of that specific probe on the healthy control organoids. For probe combinations, single organoids were counted as positive when at least one of the selected probes in the respective combination was positive. Percentage coverage was calculated by dividing the amount of positive single organoids by the total amount of organoids in the respective PDO line.

## NB xenograft model

Six-to-eight-week-old athymic nude female mice (NMRI-*Foxn1*$^{nu}$) were purchased from Charles River Laboratories. Mice were

housed under 45–65% humidity and a daily 12/12-h light/dark regime, in sterile conditions using an individually ventilated cage system and fed with low fluorophore rodent diet (D1001i, research diets) and sterile water. Mice were used for xenografting of KCNR cells and PDO lines NBL39, NBL67, and NBL129. On average, $1.3 \times 10^6$ single cells were injected subcutaneously at 2 dorsal sites in 50%medium/50%BME. Animals were anesthetized with 3–4% isoflurane for induction and 2% isoflurane for maintenance with a flow of 0.5 l/min during injection of tumor cells and imaging procedures. Tumor growth was followed up by palpation of the dorsal flanks at the tumor injection sites two times per week. When tumors were approximately $8 \times 8$ mm, mice were randomly distributed over the different dose treatment groups, injected with the FGS probes and imaging performed. Mice harboring tumors of smaller size were excluded from follow-up experiments and analysis.

Mice bearing subcutaneous tumors originating from KCNR cells were randomly distributed over the different dose treatment groups and intravenously injected in their tail vein with 0.3 nmol, 1 nmol, or 3 nmol of anti-L1CAM-IRDye800CW or anti-GD2-IRDye800CW in 50 µl PBS. Fluorescent signals were measured daily over the course of seven days using the IVIS Spectrum In Vivo Imaging System (Perkin Elmer, Waltham, MA, USA) (Fig. EV4A–D). The optimal dose and timing for both anti-L1CAM-IRDye800CW and anti-GD2-IRDye800CW was based on both the TBR and the mean fluorescence intensity (MFI). This dose of 1 nmol was used for injection of the PDO xenograft models with an optimal time of 5 days for anti-GD2-IRDye800CW and 6 days for anti-L1CAM-IRDye800CW. Fluorescent signals for all PDO xenograft models were measured as described for the KCNR xenograft model.

### In vivo TBR calculation and biodistribution

To measure the MFI on the IVIS Imaging System, regions of interest were set based on the visible tumor. IVIS imaging data were analyzed using the Living Image Software (Perkin Elmer, Waltham, MA, USA, version 4.7.4). TBRs were calculated by dividing the MFI of the tumor by the MFI of the neighboring background signal. Biodistribution of anti-GD2-IRDye800CW and anti-L1CAM-IRDye800CW was assessed by measuring the MFI of multiple organs 5 days after administration of anti-GD2-IRDye800CW and 6 days after administration of anti-L1CAM-IRDye800CW in mice injected with 1 nmol of the respective probe.

### Statistics

Statistical analyses were performed using either Prism v.9 software (GraphPad) or R (2022.12.0). After confirming normal distribution using a Shapiro–Wilk test, a two-way analysis of variance (ANOVA) was performed between in vivo and in vitro TBRs with Sidáks multiple comparison tests to adjust for multiple comparisons. All quantitative data shown are representative of at least $n = 3$ independent experiments. Sample size for mouse in vivo xenografting experiments were based on previous outcomes with the same experimental set-up (Wellens et al, 2020). No blinding was performed when analyzing data.

### The paper explained

#### Problem

Fluorescence-guided surgery (FGS) with molecular-targeted probes can improve tumor resection by assisting the surgeon to visually discriminate tumor tissue during the surgery. Finding suitable FGS probes depends on the identification of targets that are specifically overexpressed on tumor tissue, as compared to healthy tissue surrounding the tumor area. This process is challenging and can be complicated by variation in expression between individual patients, as well as different regions within the same tumor.

#### Results

To develop a screening tool for finding suitable FGS probes, we combined organoid technology with multi-spectral 3D imaging. This enables multi-colored visualization of multiple probes binding to various targets in a single acquisition, while organoids are implemented to mimic patient-derived tumor tissue in vitro. The resulting organoid-imaging platform allows to adequately capture both inter-patient and intratumoral heterogeneity in target expression. Moreover, by including healthy tissue organoids, binding to human tissue surrounding the tumor site can be evaluated, allowing to confirm the tumor-specificity of identified targets.

#### Impact

We identified potential new FGS probes for neuroblastoma based on their ability to effectively discriminate tumor tissue from healthy tissue. In addition, we designed probe combinations that could lead to more uniform detection of highly heterogenous breast cancer. Beyond FGS, our platform could be used to screen for tumor-overexpressed therapeutic targets for immunotherapy or molecular-targeted treatment.

### For more information

- RNA sequencing datasets access for screening of tumor-overexpressed membrane targets: R2 Genomics Analysis and Visualization Platform (http://r2.amc.nl).
- Protein expression dataset to verify low expression on healthy tissue of identified targets: the Human Protein Atlas (https://www.proteinatlas.org).

## Data availability

Imaging data has been deposited in BioImage Archive (https://www.ebi.ac.uk/biostudies/bioimages/studies), accession number: S-BIAD1119.

The source data of this paper are collected in the following database record: biostudies:S-SCDT-10_1038-S44321-024-00084-4.

## Peer review information

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

## Acknowledgements

We thank the Princess Máxima Center for Pediatric Oncology for technical support and the Hubrecht Institute and Zeiss for imaging support and collaboration. All imaging was performed at the Princess Máxima Imaging Center. We thank the HUB for providing BC PDOs, the Princess Máxima Center Organoid Facility for organoid culture support, and J. Drost and J. Bühl (Princess Máxima Center for Pediatric Oncology) for providing healthy kidney organoid lines. We also thank the R2 Genomics Analysis and Visualization Platform (http://r2.amc.nl), and J. Koster and R. Volckmann in particular, for RNA database availability and analyses support, Elthera for providing L1CAM antibodies, and QVQ for supplying NCAM1 nanobodies. We thank E. Bokobza and M. Buchholz for providing the healthy breast organoid lines and members of the Dream3D^LAB (Rios group) for offering critical feedback on the project and manuscript. This work was financially supported by the Princess Máxima Center for Pediatric Oncology, Oncode Institute, the Netherlands, and the Dutch Neuroblastoma Foundation (Villa Joep). ACR was supported by an European Research Council (ERC)-starting grant 2018 project (no. 804412).

## Author contributions

**Bernadette Jeremiasse**: Conceptualization; Data curation; Formal analysis; Validation; Investigation; Visualization; Methodology; Writing—original draft; Writing—review and editing. **Ravian L van Ineveld**: Formal analysis; Investigation; Visualization; Methodology; Writing—review and editing. **Veerle Bok**: Formal analysis; Investigation; Methodology. **Michiel Kleinnijenhuis**: Software; Formal analysis; Methodology. **Sam de Blank**: Software; Formal analysis; Validation; Methodology. **Maria Alieva**: Data curation; Formal analysis; Validation. **Hannah R Johnson**: Investigation; Methodology. **Esmée J van Vliet**: Formal analysis; Visualization. **Amber L Zeeman**: Investigation; Methodology. **Lianne M Wellens**: Conceptualization; Methodology. **Gerard Llibre-Palomar**: Formal analysis; Methodology. **Mario Barrera Roman**: Data curation; Formal analysis; Investigation. **Alessia Di Maggio**: Resources. **Johanna F Dekkers**: Supervision; Methodology. **Sabrina Oliveira**: Resources. **Alexander L Vahrmeijer**: Writing—review and editing. **Jan J Molenaar**: Resources. **Marc HWA Wijnen**: Supervision; Writing—review and editing. **Alida FW van der Steeg**: Supervision; Writing—review and editing. **Ellen J Wehrens**: Writing—original draft; Writing—review and editing. **Anne C Rios**: Conceptualization; Supervision; Funding acquisition; Writing—original draft; Project administration; Writing—review and editing.

Source data underlying figure panels in this paper may have individual authorship assigned. Where available, figure panel/source data authorship is listed in the following database record: biostudies:S-SCDT-10_1038-S44321-024-00084-4.

## Disclosure and competing interests statement

ACR and JFD are inventors on a pending patent related to breast organoid technology (WO201309812-A3/WO2016083612-AI/P309013GB).

# Expanded View Figures

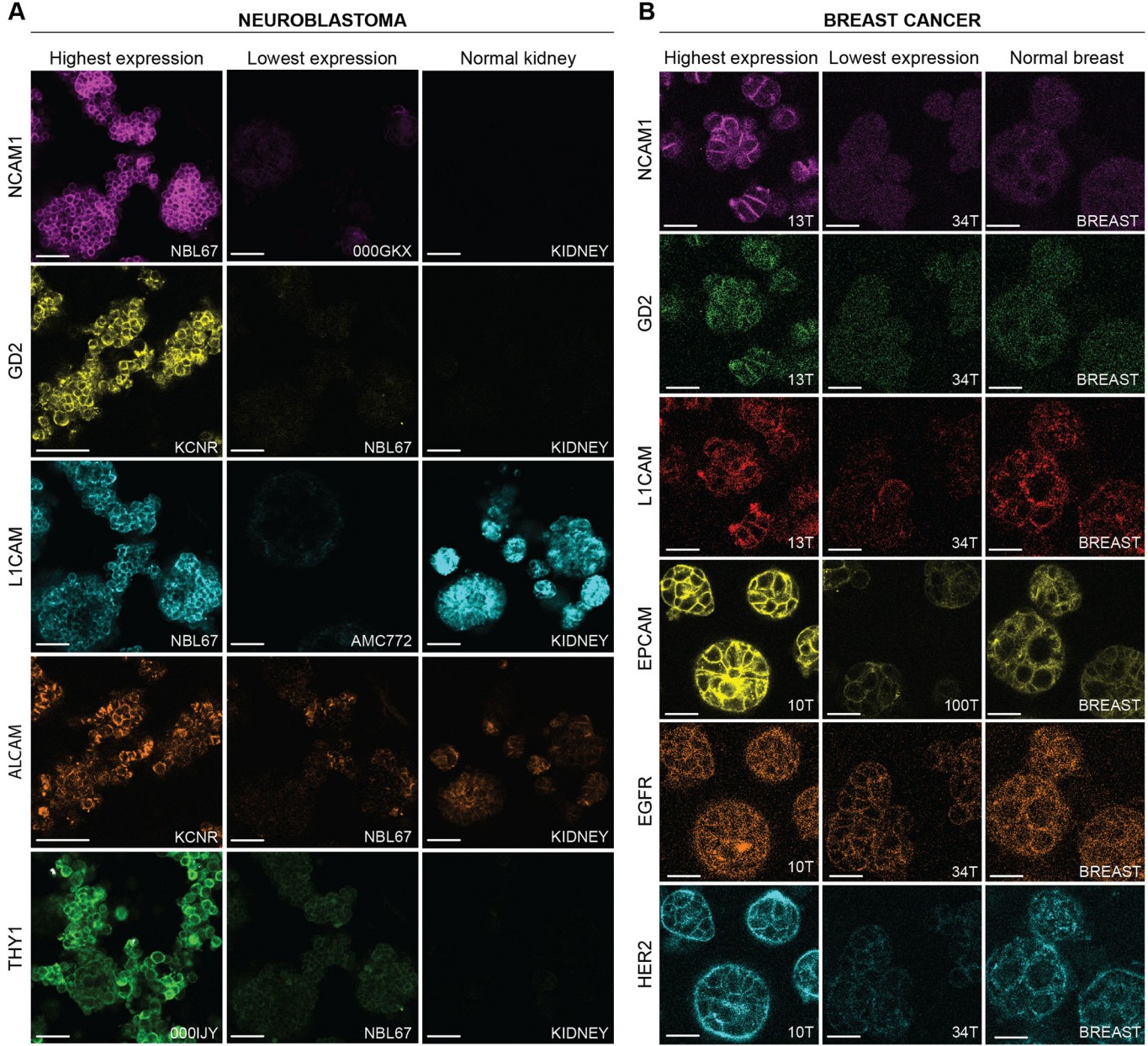

**Figure EV1. Representative single-channel optical sections.**

(A, B) Images of optical sections representing the highest (left) and lowest (middle) expression of the screened probes and expression on a healthy tissue control organoid line (right) for NB (A) and BC (B). Scale bars 50 μm (A) and 30 μm (B).

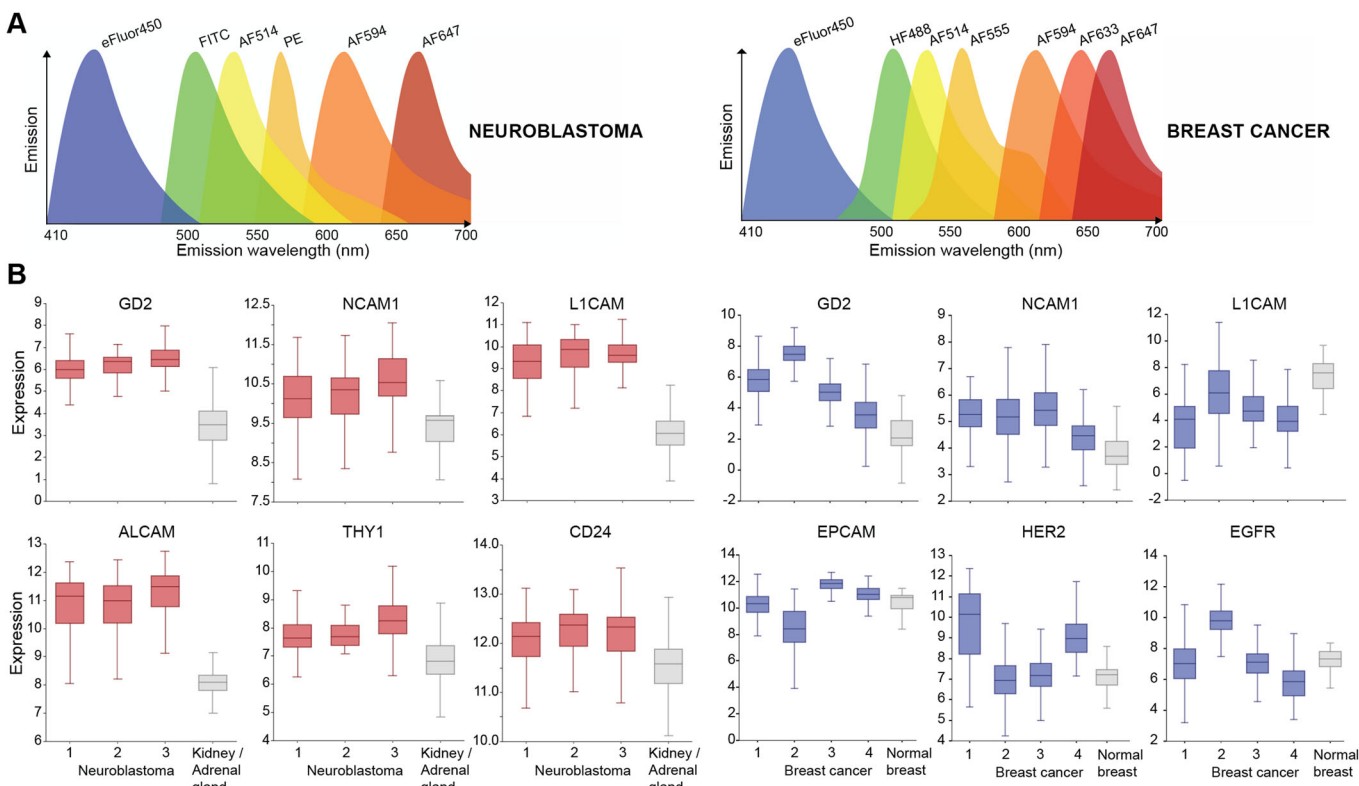

**Figure EV2. Emisson spectra and target selection.**

(A) Schematic representation of the fluorophore emission spectra for 7-color imaging of NB (left) and BC (right). (B) Box plot depicting relative target gene expression from R2 RNA sequencing datasets for NB (left) and BC (right) and respective adjacent healthy control tissue. Centre: median, bounds: Q1–Q3, whiskers extend to minimum/maximum limited to 1.5 times the IQR. Details of sequencing datasets, including sample size, is provided in Methods.

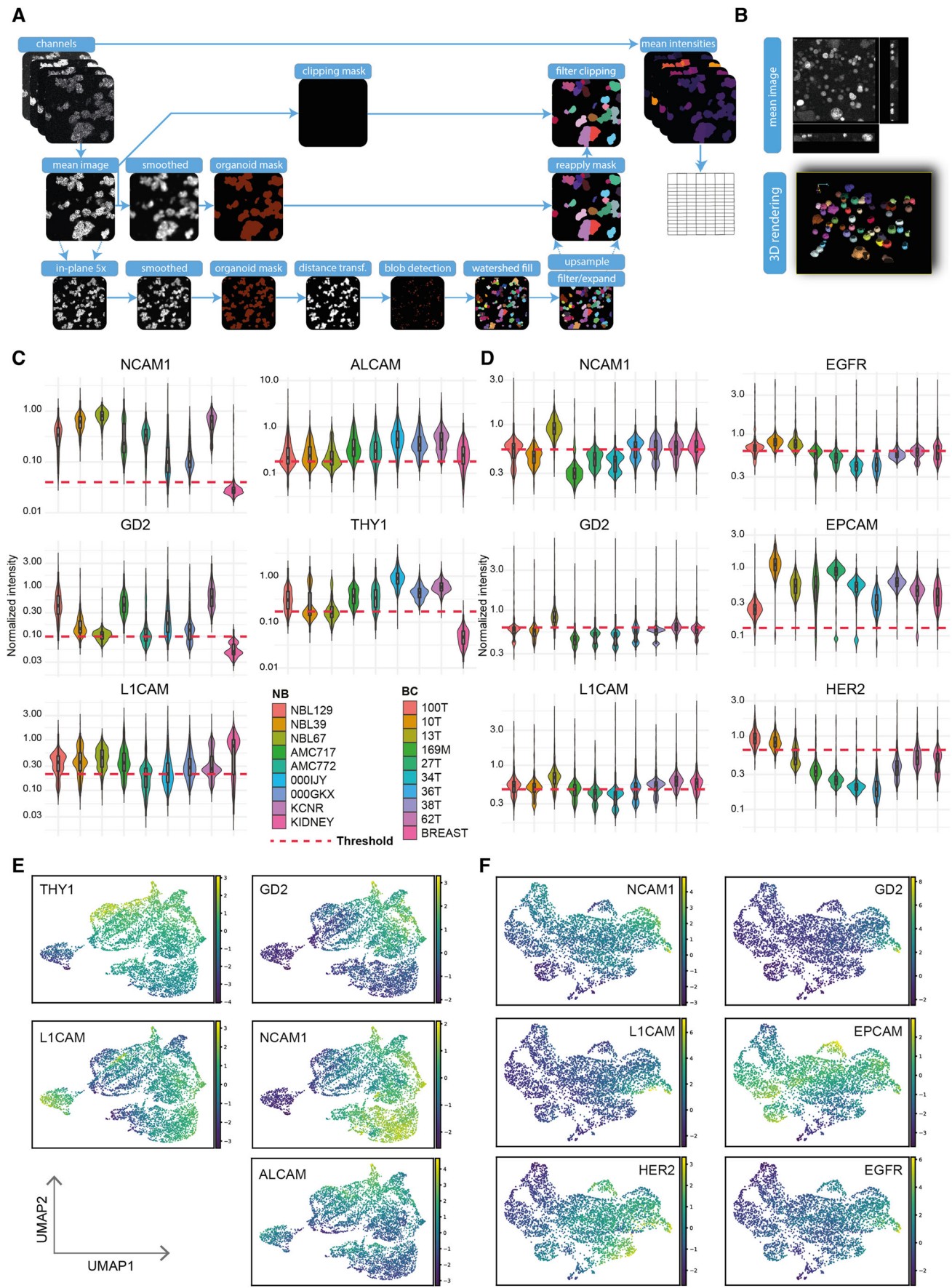

◄ **Figure EV3. STAPL-3D pipeline optimized for individual organoid segmentation and fluorescent intensity extraction.**

(A) Schematic representation of the STAPL-3D pipeline with key optimization steps for single organoid segmentation. (B) Representative raw 3D imaging data and associated 3D rendered single organoid segmentation. (C, D) Violin plots of the mean fluorescent intensities (MFI) of the screened probes for all individual organoids from the NB (C) and BC (D) PDO biobanks normalized for the experimental day and probe. Dashed line indicates the cut-off used for considering an organoid as positively stained. Boxplots inside violin plots; centre: median, bounds: Q1–Q3, whiskers extend to minimum/maximum limited to 1.5 times the IQR. $n = 3$ independent experiments. (E, F) Representation of the spatial target distribution used for UMAP clustering of both the NB (E) and BC (F) PDO biobank. $n = 3$ independent experiments.

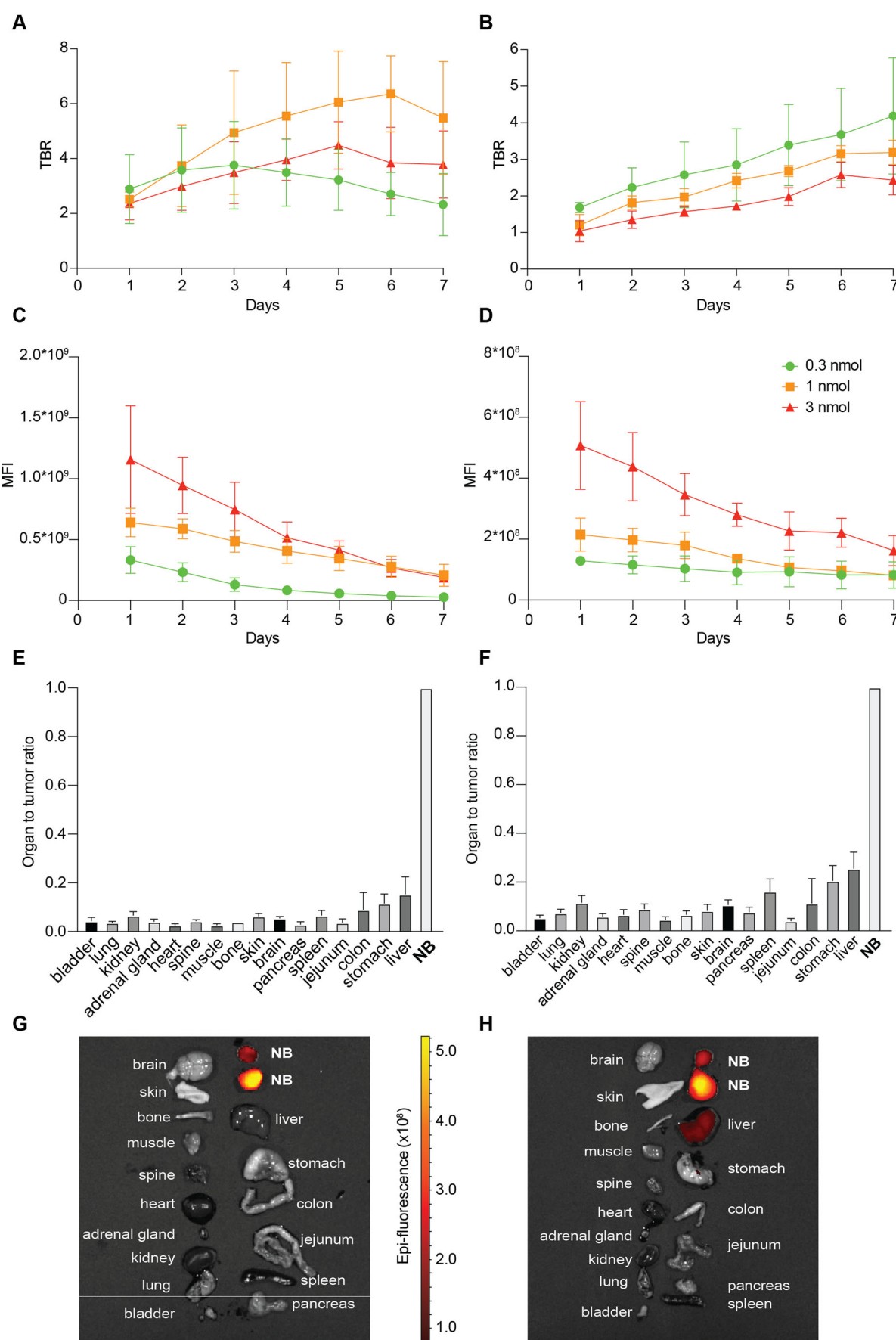

**Figure EV4.   In vivo testing of anti-GD2-IRDye800CW and anti-L1CAM-IRDye800CW in NB xenografts.**

(**A–D**) Line graph depicting the TBR (**A, B**) and MFI (**C, D**) of anti-GD2-IRDye800CW (**A, C**) and anti-L1CAM-IRDye800CW (**B, D**) per dose on 7 consecutive days. Mean TBR or MFI ± SD as imaged with the IVIS Spectrum system, $n = 3$ to 4 mice per dose group. (**E, F**) Bargraphs of the biodistribution of anti-GD2-IRDye800CW at day 5 (**E**) and anti-L1CAM-IRDye800CW at day 6 (**F**) in subcutaneous NB tumor-bearing mice receiving a 1 nmol dose. Mean MFI of organs and tissues normalized to the tumor + SD of $n = 3$ mice per probe. (**G, H**) Representative images of the biodistribution of anti-GD2-IRDye800CW at day 5 (**G**) and anti-L1CAM-IRDye800CW at day 6 (**H**) in subcutaneous NB tumor-bearing mice receiving a 1 nmol dose.

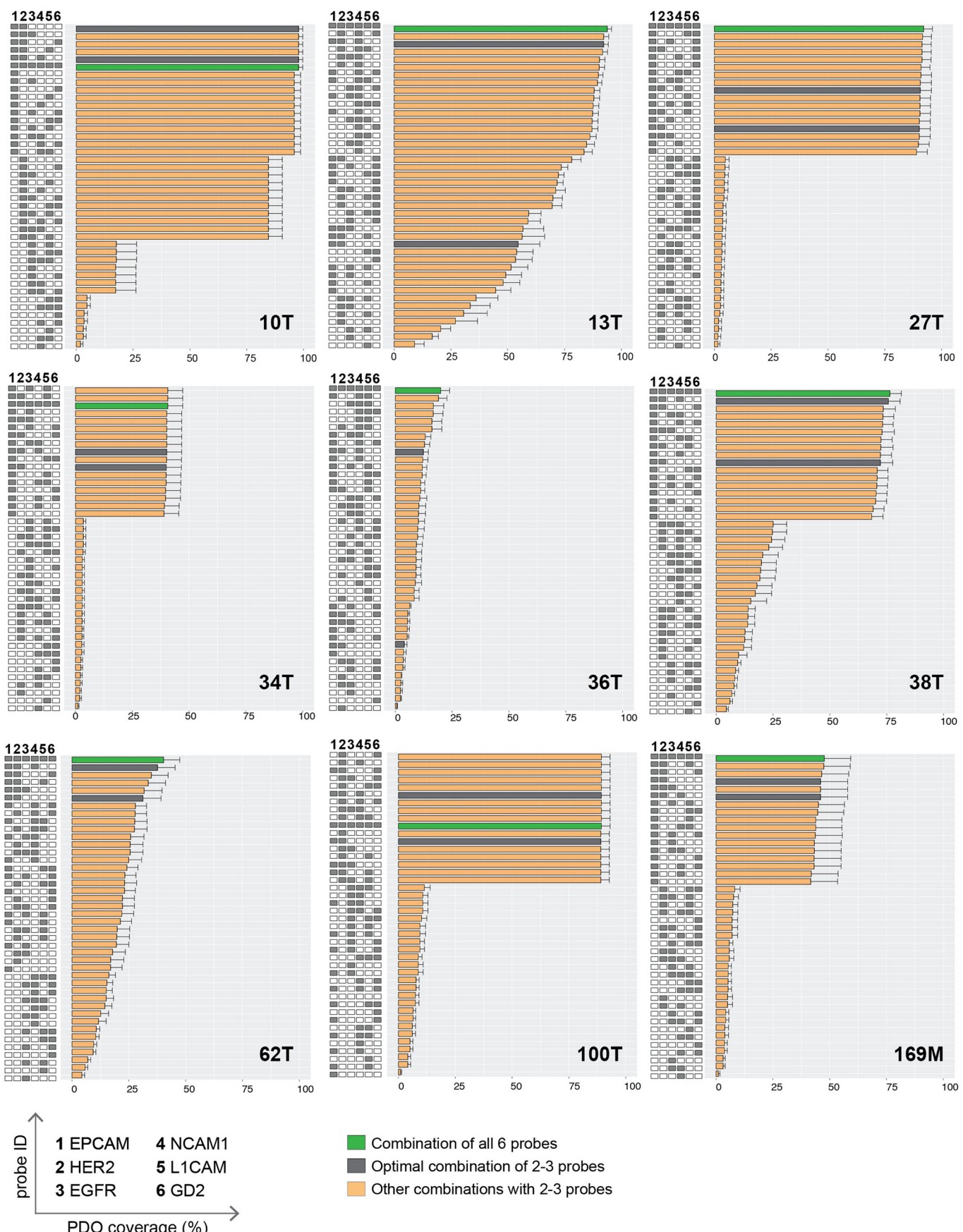

**probe ID** → **PDO coverage (%)**

- **1** EPCAM
- **2** HER2
- **3** EGFR
- **4** NCAM1
- **5** L1CAM
- **6** GD2

- 🟩 Combination of all 6 probes
- ⬛ Optimal combination of 2-3 probes
- 🟧 Other combinations with 2-3 probes

**Figure EV5. Percentage organoid coverage with single probes and probe combinations per BC PDO line.**

Percentage BC organoids that are positive for individual probes or probe combinations. Gray bars depict the 2 overall most effective probe combinations with a limited number of probes. In case of similar percentages, combinations are ranked higher if they consist of less probes. Green bars represent the maximum achievable percentage of coverage obtained with the combination of all six probes tested. Bars depict mean percentage $+$ SEM of $n = 3$ independent experiments.

