## [Peer Review File · EMBO Molecular Medicine]

A multispectral 3D live organoid imaging platform to screen probes for fluorescence guided surgery

Anne Rios, Bernadette Jeremiassé, Ravian van Ineveld, Veerle Bok, Michiel Kleinnijenhuis, Sam de Blank, Maria Alieva, Hannah Johnson, Esmée van Vliet, Amber Zeeman, Lianne Wellens, Gerard Llibre-Palomar, Mario Barrera Roman, Alessia Di Maggio, Johanna Dekkers, Sabrina Oliveira, Alexander Vahrmeijer, Jan Molenaar, Marc Wijnen, Alida van der Steeg, and Ellen Wehrens

Corresponding author: Anne Rios (a.c.rios@prinsesmaximacentrum.nl)

Review Timeline:

Submission Date:	20th Oct 23
Editorial Decision:	29th Nov 23
Revision Received:	24th Jan 24
Editorial Decision:	15th Feb 24
Revision Received:	26th Apr 24
Editorial Decision:	29th Apr 24
Revision Received:	13th May 24
Accepted:	21st May 24

Editor: Lise Roth

Transaction Report:

29th Nov 2023

Dear Dr. Rios,

Thank you for the submission of your manuscript to EMBO Molecular Medicine, and please accept my apologies for the delay in getting back to you as we were waiting for one referee report. However, given that referee #2 has not yet gotten back to us despite several chasers, and that both referees #1 and #3 provide similar recommendations, we prefer to make a decision now in order to avoid further delay in the process. Should referee #2 provide a report, we will send it to you, with the understanding that we will not ask you further reaching experiments.

As you will see from the reports below, both referees acknowledge the novelty and interest of the study, nevertheless both referees also mention the importance of in vivo validation in adequate mouse models of cancer.

Addressing the reviewers' concerns in full will be necessary for further considering the manuscript in our journal, and acceptance of the manuscript will entail a second round of review. EMBO Molecular Medicine encourages a single round of revision only and therefore, acceptance or rejection of the manuscript will depend on the completeness of your responses included in the next, final version of the manuscript. For this reason, and to save you from any frustrations in the end, I would strongly advise against returning an incomplete revision.

We are expecting your revised manuscript within three months, if you anticipate any delay, please contact us.

We require:

- 1) A .docx formatted version of the manuscript text (including legends for main figures, EV figures and tables). Please make sure that the changes are highlighted to be clearly visible.
- 2) Individual production quality figure files as .eps, .tif, .jpg (one file per figure). For guidance, download the 'Figure Guide PDF' (<https://www.embopress.org/page/journal/17574684/authorguide#figureformat>).
- 3) At EMBO Press we ask authors to provide source data for the main figures. Our source data coordinator will contact you to discuss which figure panels we would need source data for and will also provide you with helpful tips on how to upload and organize the files.
- 4) A .docx formatted letter INCLUDING the reviewers' reports and your detailed point-by-point responses to their comments. As part of the EMBO Press transparent editorial process, the point-by-point response is part of the Review Process File (RPF), which will be published alongside your paper.
- 5) A complete author checklist, which you can download from our author guidelines (<https://www.embopress.org/page/journal/17574684/authorguide#submissionofrevisions>). Please insert information in the checklist that is also reflected in the manuscript. The completed author checklist will also be part of the RPF.
- 6) It is mandatory to include a 'Data Availability' section after the Materials and Methods. Before submitting your revision, primary datasets produced in this study need to be deposited in an appropriate public database, and the accession numbers and database listed under 'Data Availability'. Please remember to provide a reviewer password if the datasets are not yet public (see <https://www.embopress.org/page/journal/17574684/authorguide#dataavailability>).

7) For data quantification: please specify the name of the statistical test used to generate error bars and P values, the number (n) of independent experiments (specify technical or biological replicates) underlying each data point and the test used to calculate p-values in each figure legend. The figure legends should contain a basic description of n, P and the test applied. Graphs must include a description of the bars and the error bars (s.d., s.e.m.). Please provide exact p values.

8) Our journal encourages inclusion of *data citations in the reference list* to directly cite datasets that were re-used and

obtained from public databases. Data citations in the article text are distinct from normal bibliographical citations and should directly link to the database records from which the data can be accessed. In the main text, data citations are formatted as follows: "Data ref: Smith et al, 2001" or "Data ref: NCBI Sequence Read Archive PRJNA342805, 2017". In the Reference list, data citations must be labeled with "[DATASET]". A data reference must provide the database name, accession number/identifiers and a resolvable link to the landing page from which the data can be accessed at the end of the reference. Further instructions are available at .

9) We replaced Supplementary Information with Expanded View (EV) Figures and Tables that are collapsible/expandable online. A maximum of 5 EV Figures can be typeset. EV Figures should be cited as 'Figure EV1, Figure EV2' etc... in the text and their respective legends should be included in the main text after the legends of regular figures.

10) The paper explained: EMBO Molecular Medicine articles are accompanied by a summary of the articles to emphasize the major findings in the paper and their medical implications for the non-specialist reader. Please provide a draft summary of your article highlighting

11) For more information: There is space at the end of each article to list relevant web links for further consultation by our readers. Could you identify some relevant ones and provide such information as well? Some examples are patient associations, relevant databases, OMIM/proteins/genes links, author's websites, etc...

12) Author contributions: CRediT has replaced the traditional author contributions section because it offers a systematic machine readable author contributions format that allows for more effective research assessment. Please remove the Authors Contributions from the manuscript and use the free text boxes beneath each contributing author's name in our system to add specific details on the author's contribution. More information is available in our guide to authors.

13) Disclosure statement and competing interests: We updated our journal's competing interests policy in January 2022 and request authors to consider both actual and perceived competing interests. Please review the policy <https://www.embopress.org/competing-interests> and update your competing interests if necessary.

14) Every published paper now includes a 'Synopsis' to further enhance discoverability. Synopses are displayed on the journal webpage and are freely accessible to all readers. They include a short stand first (maximum of 300 characters, including space) as well as 2-5 one-sentences bullet points that summarizes the paper. Please write the bullet points to summarize the key NEW findings. They should be designed to be complementary to the abstract - i.e. not repeat the same text. We encourage inclusion of key acronyms and quantitative information (maximum of 30 words / bullet point). Please use the passive voice. Please attach these in a separate file or send them by email, we will incorporate them accordingly.

15) As part of the EMBO Publications transparent editorial process initiative (see our Editorial at <http://embomolmed.embopress.org/content/2/9/329>), EMBO Molecular Medicine will publish online a Review Process File (RPF) to accompany accepted manuscripts.

In the event of acceptance, this file will be published in conjunction with your paper and will include the anonymous referee reports, your point-by-point response and all pertinent correspondence relating to the manuscript. Let us know whether you agree with the publication of the RPF and as here, if you want to remove or not any figures from it prior to publication. Please note that the Authors checklist will be published at the end of the RPF.

I look forward to receiving your revised manuscript.

Yours sincerely,

Lise Roth

***** Reviewer's comments *****

Referee #1 (Comments on Novelty/Model System for Author):

Need more in-vivo data to support FGS feasibility and outcome

Referee #1 (Remarks for Author):

Proof of concept for FGS ask for more in-vivo data including outcome data for all three models.

Great concept novel in sense of multispectral/multitarget approach , although need more in vivo data to support feasibility and outcome benefit.

Choice of markers may be limited and need to be customize to cover heterogeneity. This study reflects the organoid imaging in details as proof of concept to address heterogeneity although not thoroughly tested in animal model. Furthermore outcome data form Mouse model for FGS recommended.

Referee #3 (Comments on Novelty/Model System for Author):

The authors should confirm their findings in a mouse model of neuroblastoma and breast cancer metastasis, in order to check whether their probes are able to detect metastasis in vivo. This would add novelty and could have a huge impact on the field.

Referee #3 (Remarks for Author):

In the manuscript "A multispectral 3D live organoid imaging platform to screen probes for fluorescence guided surgery" the authors have tried to tackle an important problem in cancer biology, achieving complete tumor resection. In particular, they have generated new tools for fluorescence-guided surgery, an organoid-based multiplex imaging platform to screen for FGS probes. The manuscript is well written, the methodology is robust and the data are supported by the experiments. The patient-derived organoid (Neuroblastoma, Breast Cancer) approach is novel and relevant for screening multiple probes at the patient scale. However, the authors validated their platform only with neuroblastoma organoids in vivo and with subcutaneous injection, which is not a relevant site of neuroblastoma formation. Indeed, the major challenge from a clinical point of view for patients with neuroblastoma and breast cancer is metastasis formation (in bone, liver and brain).

Major Points:

1) The authors should confirm their findings in a mouse model of neuroblastoma and breast cancer metastasis, in order to check whether their probes are able to detect metastasis in vivo (i.e. tail vein injection of cells from the organoids). This approach will add novelty and could have a huge impact on the cancer field.

2) In Extended Data Fig.5g there is maybe a signal in the liver of the mouse. Is this due to a possible tumor spreading in a few days?

Point-by-point response to the reviewers' comments Jeremiassé *et al.* [EM-2023-18863]

We are very pleased by the overall positive feedback on the novelty of our approach that combines an *in vitro* organoid-based imaging assay and associated computational framework to deliver a fluorescence-guided surgery (FGS) probe screening platform to: 1) accurately quantify efficiency of labeling; 2) score tumor heterogeneity in spatial distribution and 3) define background caused by human healthy tissue. Together, this provides the advantage of comparing up to 7 FGS probes simultaneously, offering throughput to identify the most discriminative probe for a certain tumor indication and design potential probe combinations for more heterogeneous tumor types. We have addressed each reviewer's comment in detail in the point-by-point response below, as well as in the revised version of our manuscript (pg. 3; lines 63-66, pg. 6; lines 147-150 and pg. 7; lines 150-152), to better clarify the main goal of this study and address their feedback concerning FGS outcomes and metastatic disease detection. We hope that our revised manuscript, along with our responses to the reviewers, now positions our study as suitable for publication in EMBO Molecular Medicine.

Referee #1 (Comments on Novelty/Model System for Author):

Need more in-vivo data to support FGS feasibility and outcome

Referee #1 (Remarks for Author):

Proof of concept for FGS ask for more in-vivo data including outcome data for all three models.

Great concept novel in sense of multispectral/multitarget approach, although need more in vivo data to support feasibility and outcome benefit.

Choice of markers may be limited and need to be customized to cover heterogeneity. This study reflects the organoid imaging in details as proof of concept to address heterogeneity although not thoroughly tested in animal model. Furthermore outcome data from Mouse model for FGS recommended.

We thank the reviewer for the valuable feedback and thoughtful consideration of our manuscript. We are pleased with the overall positive evaluation regarding the innovative nature of our *in vitro* organoid-based platform, specifically designed for screening multiple FGS probes through a multispectral/multitarget 3D live fluorescence imaging approach. We fully agree with, and recognize, the significance of supporting our *in vitro* data with comprehensive *in vivo* data and we have conducted an extensive array of *in vivo* experiments (involving 120 animals; 40 animals used for evaluating NB organoid engraftment and 80 animals used for optimal probe dosage evaluation on 7 consecutive days and labelling assessment and *in vitro* comparison at optimal dosage (Fig 2 A-D; Fig. EV5) to address this). This substantial body of work has yielded a robust dataset that validates our proof of concept with correlative labeling efficiency observed in an *in vivo* setting. However, importantly, it also highlights differences arising from labeling background in healthy human tissue, a parameter that cannot be accurately measured in animal models, due to human specificity of the antibodies (and absence of cross-species reactivity) (see Fig 2 A-D and Fig. EV5 for reference). Therefore, we aimed to develop a tool that accurately measures labeling efficiency of

clinically relevant probes (i.e. human specific) within human cancer specimens, using patient-derived organoids. This approach has the advantage of informing surgeons not only about the labeling properties of FGS probes in cancerous tissue, but also in healthy tissue. This is critical, as only highly discriminative probes (i.e. high binding to tumor tissue and low or absent binding to healthy tissue) will be able to guide the surgeon. Moreover, we introduced a computational tool enabling the scoring of spatial distribution and tumor heterogeneity for up to 7 probes simultaneously, allowing for comparison and potential combination design to offer full tumor coverage. We, therefore, believe that this novel technology provides crucial information on labeling efficacy, tumor heterogeneity in spatial distribution, and tumor/healthy background signal in a human setting, complementing traditional animal models currently in use for FGS probe assessment. To further clarify this main objective of our work and subsequent contribution to the field, we now discuss this in the revised version of our manuscript (pg. 6; lines 147-150).

Acknowledging the importance of obtaining data on outcome benefits, we direct the reviewer to our previous study by Wellens *et al.* (Scientific Reports, 2020; see also attached video), where we demonstrated the feasibility of removing fluorescently labeled tumors *in vivo* using the molecular FGS probe; anti-GD2-IRDye800CW. However, predicting improved outcomes with FGS remains challenging. Phase III clinical trials are presently underway to substantiate the patient benefits derived from tumor-targeted FGS (Mieog *et al.*, Nature Reviews Clinical Oncology, 2022). This challenge is attributed to a multitude of actors, including tumor characteristics, probe properties, disease advancement, and patient responses. We address this need for in-patient trials to evaluate the outcome benefits of pre-clinically identified probes in the revised version of our manuscript (pg. 7; lines 150-152). We assert that comprehensively evaluating the nuanced nature of FGS outcomes cannot be solely accomplished through the use of animal models. Hence, for the reviewer's information, a clinical trial involving the FGS probe anti-GD2-IRDye800CW in a cohort of patients with neuroblastoma is scheduled to commence this year at our center (PS21DIN). Our objective is to offer comprehensive insights into outcomes and the behavior of the probe in patients in the forthcoming years.

Referee #3 (Comments on Novelty/Model System for Author):

The authors should confirm their findings in a mouse model of neuroblastoma and breast cancer metastasis, in order to check whether their probes are able to detect metastasis *in vivo*. This would add novelty and could have a huge impact on the field.

Referee #3 (Remarks for Author):

In the manuscript "A multispectral 3D live organoid imaging platform to screen probes for fluorescence guided surgery" the authors have tried to tackle an important problem in cancer biology, achieving complete tumor resection. In particular, they have generated new tools for fluorescence-guided surgery, an organoid-based multiplex imaging platform to screen for FGS probes. The manuscript is well written, the methodology is robust and the data are supported by the experiments. The patient-derived organoid (Neuroblastoma, Breast Cancer) approach is novel and relevant for screening multiple probes at the patient scale. However, the authors validated their platform only with neuroblastoma organoids in

vivo and with subcutaneous injection, which is not a relevant site of neuroblastoma formation. Indeed, the major challenge from a clinical point of view for patients with neuroblastoma and breast cancer is metastasis formation (in bone, liver and brain).

We are very pleased that the reviewer found our study well-designed and experimentally sound and has positively evaluated the innovative aspect of our *in vitro* organoid/imaging screening platform. While we fully agree with the reviewer's observation that the most significant clinical challenge in cancer treatment involves addressing highly progressed and metastasized disease, it is important to note that, at present, FGS, aside from metastatic sentinel lymph nodes and peritoneal metastases (Subramanyeshwar et al., Journal of Surgical Oncology, 2021), is predominantly utilized for bulk tumor resection rather than the treatment of metastatic disease. To better clarify our intention to the reviewer: our study's primary objective was to establish a robust *in vitro* organoid-based platform. This platform is specifically designed for the screening of potential FGS probes by employing specimens derived directly from patients.

Major Points:

1) The authors should confirm their findings in a mouse model of neuroblastoma and breast cancer metastasis, in order to check whether their probes are able to detect metastasis *in vivo* (i.e. tail vein injection of cells from the organoids). This approach will add novelty and could have a huge impact on the cancer field.

While for now FGS is predominantly applied for tumor resection/debulking and not metastasis detection or resection, this insightful observation has prompted us to further reflect on the clinical implications of our study. As mentioned above, the primary aim of our research was to establish a robust *in vitro* organoid-based platform for screening potential FGS probes using patient-derived specimens. One important advantage of our approach is the inclusion of healthy human-derived organoids alongside cancerous ones, enabling the quantification of potential background in healthy versus cancerous human-derived tissues. The primary challenge for achieving successful FGS for tumor resection lies in a strong specific expression on tumor target tissue relative to healthy surrounding tissue (Sutton et al., BJS Open, 2023). This is particularly crucial because these healthy tissues often exhibit similar marker expression, making the discrimination more complex. However, in the breast cancer (BC) organoid panel, we incorporated a patient-derived line from metastatic tissue. This showcases that our *in vitro* platform could in the future also be used for defining molecular probes relevant to metastasis detection, and we discuss this future application in the revised version of our manuscript on pg. 3 (lines 63-66). Further details on this specific metastatic BC line (169M) can be found in Table EV1 of our manuscript, and additional information is available in Dekkers et al., Nature Protocols, 2021, where we extensively describe the BC biobank, including the lines used in the presented work (see Table 1 of this paper; HUB-01-C2-152 for the 169M metastatic BC PDO line). However, this requires investigation of different criteria compared to those required for tumor resection (the primary objective of our current work), i.e. risk of background binding to healthy tissue will be less, as metastases per definition are found in different organs compared to the original tumor, and binding and resulting fluorescence need to be strong enough to detect small micrometastases.

2) In Extended Data Fig.5g there is maybe a signal in the liver of the mouse. Is this due to a possible tumor spreading in a few days?

We thank the reviewer for this valuable comment concerning Fig. EV5G. The signal detected in the liver corresponds to the biodistribution of anti-GD2-IRDye800CW, and we attribute this signal to the degradation of the probe within the liver. The liver is a site known to exhibit significant fluorescence *in vivo*, due to the degradation of probes, and this phenomenon has been extensively documented in the literature in various studies exploring the biodistribution and degradation patterns of fluorescent probes in different tissues. Specifically, in our previous work by Wellens *et al.* (Scientific Reports, 2020), we observed a similar pattern, where we extensively studied GD2 labeling for *in vivo* intraoperative imaging using xenograft neuroblastoma models. The findings from this prior work align with the observations in the current study.

15th Feb 2024

Dear Anne,

Thank you for submitting your revised manuscript. We have now received the report from the 2 referees who re-reviewed your manuscript. As you will see below, while referee #1 is satisfied with the explanations provided, referee #3 still thinks that in vivo validation is warranted. We therefore additionally consulted an external advisor, who stated:

"I agree that the main aspect is the platform. [...] In other words, applications are many and they should be tested afterwards. Having said that I would of course be interested in seeing how they apply to a more in vivo setting, such as brain tumors (including metastases), where 5-ALA has been used but never established because of the heterogeneity in the interpretation of its use."

Therefore, having discussed your manuscript, the referees' reports, and the expert advice one more time within the team, we agree that the point of your short report is to demonstrate the feasibility of using an organoid platform to inform fluorescence guided surgery, and that additional validation experiments will not be needed. We would however ask you to further discuss the limitations/perspectives of your work in a revised version of your manuscript.

Additionally, please address the following editorial concerns:

1/ Authors: please make sure that the email addresses from the following co-authors are correct (emails bounced back):

Veerle Bok (V.L.H.Bok@prinsesmaximacentrum.nl),
Lianne Wellens (L.M.Wellens@prinsesmaximacentrum.nl),
Michiel Kleinnijenhuis (M.Kleinnijenhuis@prinsesmaximacentrum.nl)

2/ Manuscript text:

- Please remove the blue text, and only keep in track changes mode any new modification.
- The main manuscript should contain the following sections in this order: Abstract, Keywords, The Paper Explained, Introduction, Results, Discussion, Materials and Methods, Acknowledgements, Disclosure and competing interests statement, For More Information, References, Figure legends, (main) Tables and their legends.
- Please include the Materials and Methods section, currently in supplementary information, in the main manuscript text. Information provided in the manuscript should also be included in the authors' checklist (see below):
 - o Please include a statement that the experiments conformed to the principles set out in the WMA Declaration of Helsinki and the Department of Health and Human Services Belmont Report.
 - o Please make sure to indicate the origin of all cell lines, and mention whether they were authenticated.
 - o Statistics: Please provide information on randomization, blinding, inclusion/exclusion criteria, sample size.
- Data availability: It is mandatory to include a 'Data Availability' section after the Materials and Methods. Before submitting your revision, primary datasets produced in this study need to be deposited in an appropriate public database, and the accession numbers and database listed under 'Data Availability'. In case you have no data that requires deposition in a public database, please state so in this section ("This study includes no data deposited in external repositories"). Note that the Data Availability Section is restricted to new primary data that are part of this study.
- Acknowledgements: please make sure that the funding information provided in the manuscript matches the information provided in the submission system (currently Princes Máxima Center for Pediatric Oncology is missing in the submission system).
- Author contributions: CRediT has replaced the traditional author contributions section because it offers a systematic machine-readable author contributions format that allows for more effective research assessment. Please remove the Authors Contributions from the manuscript and use the free text boxes beneath each contributing author's name in our system to add specific details on the author's contribution. More information is available in our guide to authors.
- Disclosure statement and competing interests: We updated our journal's competing interests policy in January 2022 and request authors to consider both actual and perceived competing interests. Please review the policy <https://www.embopress.org/competing-interests>, rename this section and update your competing interests if necessary.
- References: please format the references to have them in alphabetical order, with 10 authors before et al. DOIs should be removed from the published references.

3/ Figures and Appendix:

- Please provide individual production quality figure files as .eps, .tif, .jpg (one file per figure).
- Your manuscript currently contains 2 main figures and 6 EV figures. We usually allow a maximum of 5 EV figures. Would you consider making one or more of your EV figures main figures?
- Please provide exact p values, not a range, in the figures or their legends, including for non significant (ns).
- I understand that the movie will not be part of the published manuscript. If you wish to submit a movie, it should be renamed Movie EV1, and needs a legend zipped to the movie file.
- The supporting information should be removed and merged with the main manuscript text. EV tables currently included in the supporting information should be uploaded as individual files.

- Please make sure that main and EV figures are called out in chronological order (Table EV4 is called out before Table EV3).
- Figure/panel re-use must be mentioned in the figure legends (i.e. Figure 1B and Figure 2B).
- Our data editors are currently working on your manuscript. We will send you their comments in a couple of days, please address them in the figure legends.

4/ At EMBO Press we ask authors to provide source data for the main figures. Please find attached to this email the Source Data checklist, with the requests from our source data coordinator.

5/ Please provide a complete author checklist, which you can download from our author guidelines (<https://www.embopress.org/page/journal/17574684/authorguide#submissionofrevisions>). Please insert information in the checklist that is also reflected in the manuscript. The completed author checklist will also be part of the RPF.

6/ The paper explained: EMBO Molecular Medicine articles are accompanied by a summary of the articles to emphasize the major findings in the paper and their medical implications for the non-specialist reader. Please provide a draft summary of your article highlighting

7/ Every published paper now includes a 'Synopsis' to further enhance discoverability. Synopses are displayed on the journal webpage and are freely accessible to all readers. They include a short stand first (maximum of 300 characters, including space) as well as 2-5 one-sentences bullet points that summarizes the paper. Please write the bullet points to summarize the key NEW findings. They should be designed to be complementary to the abstract - i.e. not repeat the same text. We encourage inclusion of key acronyms and quantitative information (maximum of 30 words / bullet point). Please use the passive voice. Please attach these in a separate file or send them by email, we will incorporate them accordingly.

8/ As part of the EMBO Publications transparent editorial process initiative (see our Editorial at <http://embomolmed.embopress.org/content/2/9/329>), EMBO Molecular Medicine will publish online a Review Process File (RPF) to accompany accepted manuscripts.

This file will be published in conjunction with your paper and will include the anonymous referee reports, your point-by-point response and all pertinent correspondence relating to the manuscript. Let us know whether you agree with the publication of the RPF and as here, if you want to remove or not any figures from it prior to publication.

I look forward to receiving your revised manuscript.

With kind regards,

Lise

**** Reviewer's comments ****

Referee #1 (Remarks for Author):

Is suitable for publication

Referee #3 (Comments on Novelty/Model System for Author):

Subcutaneous injection is not clinically/experimentally relevant. Their technology could not be functional on breast tissue in vivo, the problem is not to detect the big tumors, but the infiltrated cells that remain after surgery.

The authors should focus on bulk tumor resection on relevant breast cancer models with surgery in the breast. I suggest testing their technology also in relevant mouse models of neuroblastoma and breast cancer metastasis.

Referee #3 (Remarks for Author):

Subcutaneous injection is not clinically/experimentally relevant. The authors validated their platform only with neuroblastoma organoids in vivo and with subcutaneous injection, which is not a relevant site of neuroblastoma formation. Indeed, the major challenge from a clinical point of view for patients with neuroblastoma and breast cancer is metastasis formation (in bone, liver and brain).

The authors should focus on bulk tumor resection on relevant breast cancer models with surgery in the breast. I suggest testing their technology also in relevant mouse models of neuroblastoma and breast cancer metastasis.

The authors addressed the editorial issues.

29th Apr 2024

Dear Anne,

Thank you for submitting your revised files. Before I can accept your manuscript, please address the following editorial concerns:

1. Please provide a rebuttal letter to the last decision and/or include a discussion on the limitations/perspectives of your work in the manuscript.
2. We think the category "Method" would be better suited to your manuscript than "Report". If you agree with changing the article category, please follow the following guidelines:
<https://www.embopress.org/page/journal/17574684/authorguide#structuredmethods>
3. Data availability: please remove the sentence "Numerical data are provided as linked data source files" and provide a URL for the data deposited on Biolmage Archive.
4. Please address the queries from our data editors:
 - a. Please note that a separate 'Data Information' section is required in the legends of figures 2a-g; EV 3c-f; EV 4e-f.
 - b. Please note that the box plots need to be defined in terms of minima, maxima, centre, bounds of box and whiskers, and percentile in the legends of figures EV 2b; EV 3c-d.
 - c. Please note that information related to n is missing in the legend of figure EV 2b.
 - d. Please note that the measure of center for the error bars needs to be defined in the legends of figures EV 4e-f.
5. Checklist: In the first section, 'Newly created materials', do any restrictions apply: could you please explain what you are referring to?
6. Thank you for providing a beautiful synopsis picture. I discussed with my colleagues, and we would like to feature your image on our cover (free of charge). Would you be interested in this option? If so, my colleague Zeljko Durdevic will contact you with further information. Regarding the synopsis, we would then encourage you to provide a graphical abstract instead (schematic that summarizes the paper).

Please let us know if you agree with the publication of the Review Process File (RPF) (that include the anonymous referee reports, your point-by-point response and all pertinent correspondence relating to the manuscript) alongside your manuscript.

I look forward to receiving your revised manuscript at your earliest convenience.

With kind regards,

Lise

Dear Lise,

Thank you for following up with additional editorial instructions for our accepted manuscript. We now submitted a further revised version of our paper (EMM-2023-18863-V4) that incorporates all additionally received editorial feedback, including a more expanded discussion on the perspective and limitations of our work.

In addition, we are definitively interested in providing our image for the journal cover and have now uploaded both a high quality tif file of this cover image, as well as a graphical abstract to accompany the synopsis of our paper.

Since we 1) identify new probe targets for fluorescence guided surgery of neuroblastoma, 2) suggest probe combinations for breast cancer, and 3) demonstrate a critical need to model healthy tissue binding, we feel publishing our work as a Method would not underscore these findings that we obtained with our platform. Therefore, we did not reformat our manuscript to a Method paper.

Finally, we agree with the publication of the Review Process File and already uploaded a formatted text file with our point-by-point responses to the reviewers' comments with previous versions of the paper.

We hope that with these additional changes, we now meet all the publication requirements.

Best,

Anne

21st May 2024

Dear Anne,

Thank you for sending the revised files. I am pleased to inform you that your manuscript is accepted for publication and is now being sent to our publisher to be included in the next available issue of EMBO Molecular Medicine!

If you have any questions, please do not hesitate to contact the Editorial Office.
Thank you for your contribution to EMBO Molecular Medicine!

With kind regards,

Lise
